# What Difference Does Public Participation Make? An Alternative Futures Assessment Based on the Development Preferences for Cultural Landscape Corridor Planning in the Silk Roads Area, China

**Haiyun Xu [1],\*, Tobias Plieninger [2,3] , Guohan Zhao [1] and Jørgen Primdahl [1]**

[1]  Department of Geosciences and Natural Resource Management, University of Copenhagen, 1958 Frederiksberg, Denmark; gz@ign.ku.dk (G.Z.); jpr@ign.ku.dk (J.P.)

[2]  Faculty of Organic Agricultural Sciences, University of Kassel, 37213 Witzenhausen, Germany; plieninger@uni-kassel.de

[3]  Department of Agricultural Economics and Rural Development, University of Göttingen, 37073 Göttingen, Germany

\*  Correspondence: hx@ign.ku.dk

**Abstract:** Landscape corridor planning (LCP) has become a widespread practice for promoting sustainable regional development. This highly complex planning process covers many policy and planning issues concerning the local landscape, and ideally involves the people who live in the area to be developed. In China, regional planners and administrators encourage the development of landscape corridor planning. However, the current LCP process rarely considers ideas from local residents, and public participation is not recognized as beneficial to planning outcomes. We use a specific Chinese case of LCP to analyze how citizen involvement may enrich sustainable spatial planning in respect to ideas considered and solutions developed. To this end, we compare a recently approved landscape corridor plan that was created without public participation with alternative solutions for the same landscape corridor, developed with the involvement of local residents. These alternatives were then evaluated by professional planners who had been involved in the initial planning process. We demonstrate concrete differences between planning solutions developed with and without public participation. Further, we show that collaborative processes can minimize spatial conflicts. Finally, we demonstrate that public participation does indeed contribute to innovations that could enrich the corridor plan that had been produced exclusively by the decision-makers. The paper closes with a discussion of difficulties that might accompany the involvement of local residents during sustainable LCP in China.

**Keywords:** cultural landscape corridor planning; participation; conflicts; development preferences; alternative future assessments; scenario planning

## 1. Introduction

Landscape corridor planning (LCP) has become a widespread practice in recent decades, partly driven by a collective shift toward protecting against fragmentation caused by infrastructure and other urban and agricultural development, and partly as a way to promote regional sustainable development and to enhance ecological and cultural values. LCP typically includes traditional approaches to landscape planning including surveys and an analysis of spatial patterns, functions, and changes, combined with various forms of stakeholder involvement in the plan-making process [1]. Throughout the paper, LCP is understood to be a form of spatial planning through which future developments

of the landscape corridor in question are both envisioned and controlled through different measures. Land use regulations, management incentives, habitat restoration and tourist facility investments are examples of such measures. In the next section, we discuss the field of planning with reference to a traditional definition of planning and a more strategic form of spatial planning.

Landscape corridors, or "greenways" [2], are linear patterns that provide connectivity across landscapes and regions and are thus subject to public spatial planning processes. LCP represents a complex form of planning that includes an array of policy and planning issues and involves many types of expertise decision-makers [3,4]. Whereas early approaches to landscape corridor planning were based almost exclusively on landscape analysis and judgements made by planning experts [5,6], these planning processes have become common practice in many countries.

Although the value and role of public participation in spatial planning have been discussed for a long time, the development and the current state of the art in terms of participatory approaches to planning vary significantly from country to country. In the West, there is a long tradition for public participation, whereas developing countries like China, participation represents a relatively new dimension of planning.

For instance, as early as the 1960s, Davidoff claimed that an inclusive planning process encourages a more democratic form of urban planning and management [7]. Since then, planning scholars have emphasized the increasing importance of public participation during regional plan-making. Over time, public participation came to be regarded as a means of reflecting democratic ideals within local planning and development [8–10], and multiple case studies have since indicated that broader public participation and collaboration can improve the process of planning and managing landscapes [11,12]. Despite these benefits, there are to our knowledge no studies of the concrete differences in planning solutions between planning with or without participation to indicate potential enrichments through participation, and there surely are no such studies with reference to China. From a planning solution point of view, participation appears to under-researched.

Despite this scholarly shortcoming, there been a long-standing debate about the value of public participation, including whether it is necessary or worthwhile to incorporate public input into decision-making [13]. For example, the process of collecting local resident feedback requires more resources because municipalities have to satisfy more stakeholders, resulting in more lengthy and costly planning and construction processes [14], or because the collaborative process sometimes could be deliberately designed to slow down environmental decision-making to favour the status quo instead of promoting the new project [15].

On this background, we will be focusing on how public participation may contribute to planning solutions in LCP using one specific case study: the landscape corridor plan in Zhangye Municipality located within the historic Silk Roads region in Western China. Our focus was the potential for the participation of local residents to enrich the overall planning content.

Landscape corridor planning in China has received increased policy attention, especially after the Silk Roads areas and the Great Canal were included as cultural routes in the list of UNESCO World Heritage Sites [16,17]. Several local and regional corridors have been or currently are being planned along these sites, and the typical planning approach is mainly based on expert analysis and judgement, supplemented only with the preferences of high-level municipal decision-makers. Here, public participation in the planning process has not been recognized as beneficial to planning outcomes in LCP.

Given this background, and with reference to the Zhangye planning case, we thus addressed the following questions: (1) To what extent does the planning content related to land use zoning and proposed major landscape projects differ from local stakeholder development preferences? and (2) What differences would have been made to the planning content if the planning process had included participation from local residents? (3) How do professional planners involved in the current plan perceive alternative plan solutions based on after-the-fact inputs from local residents?

## 2. Landscape Corridor Planning—Approaches to Analysis and Public Participation

Upon review of the dominant definitions of planning in the 1970s, Lundquist [18] suggested the following (translated from Swedish): "Planning is a future-oriented process through which the actor seeks control over the environment in order to be able to pursue his intentions." This definition is about pursuing the intentions of an "actor", and gaining control to make the future more certain. Landscape planning at that time was about surveying and analyzing the conditions and potential of a given landscape based on proposed planning solutions, which basically were about protecting what should not be changed, and locating new developments based on the best available areas and sites outside these protected areas. In Ian McHarg's book Design with Nature, such an approach formed the backbone of these highly innovative examples, showing how to integrate ecological dimensions into spatial planning for different types of landscapes and planning problems, and on different scales [5].

Since those early interpretations, this approach to spatial planning developed further in Europe and North America and has become a practice that includes stakeholders of various kinds. There are several reasons for this, including some related to politics and economics, but the bottom line is that over time, it has simply become increasingly clear that spatial planning and subsequent implementation does not function well without the involvement of key stakeholders [19,20]. Collaborations, co-design, and co-creations have become common terms for such approaches. According to Healey [21], strategic spatial planning may be understood as "a self-conscious collective effort to re-imagine a city, urban region or wider territory, and to translate the result into priorities for the area and strategic infrastructure investments, conservation measures, and principles of land use regulation."

Public participation and stakeholder involvement has also gained importance in landscape planning [22], including LCP [2]. It may, however, still be an exaggeration to claim that public participation is mainstream in LCP practice. In a review of LCP cases in Europe, it was found that public stakeholders and their approaches to involving public participation were rarely considered in current landscape corridor planning practices [1]. In traditional landscape planning, including LCP, various surveys and analyses of the landscape in question are carried out to support the implementation of the plan. Such work includes historical map-based analyses, sustainability analyses of various kinds, ecological analyses, several analyses of climate conditions, visual analyses, and many others [23]. Turner [24,25] has, for instance, criticized this Survey-Analysis-Design approach for being too mechanistic, and for being based on the assumption that, if done properly, such surveys and analyses may, more or less automatically, lead to complete planning solutions. Turner terms it the SAD approach because it often leads to "sad results". Stiles [26], on the other hand, disagrees with this view, arguing instead that the SAD approach should not be abandoned because, despite its limitations, it nonetheless supports rational solutions and remains open to criticism.

Whereas it is difficult to imagine that an LCP process can be carried out without surveys and analyses, we can nevertheless agree with Turner that in practice, the SAD method may not be sufficient to deal efficiently or effectively with most landscape planning problems. It is basically based on expert judgements alone and, as such, may hamper collaboration on priority tasks and in creating design solutions.

## 3. Public Participation in Planning Process in China

Spatial planning in China has developed mainly as an expert-driven technical process that focuses on social, economic, and environment objectives formulated in advance by governmental bodies, as well as on the physical organization of space as understood and designed mainly by professional planners [27]. Even though balancing different demands of public and private interests is a key element in spatial planning process [28], the current top-down planning process in China has a large effect on which public interests are given a voice in spatial planning, including LCP.

In China, although public participation in spatial planning is far from widespread, it is evolving, and a fast-growing body of literature arguing for more participation in Chinese planning is emerging. Still, more research about the effects of community participation, public hearings, social impact

assessments, and user discussions in regional sustainable development are needed [29–31], as Chinese political and economic systems are different from some modelled Western capitalist systems. It is therefore likely that including public involvement in Chinese landscape planning practices will evolve in other ways. These differences are due largely to weakly developed state-civil society relationships and environmental legislation [30].

According to Van der Ploeg, ignoring local resident interests on local resources and land use can lead to conflicts around rural development in China [32]. Conflicts in the past decades have been mainly due to the lack of a clear set of common interests among the highly diverse populations of rural China [33]. These conflicts are thus caused by diverse interests among differing groups of people, a classic situation in planning everywhere in the world. For instance, residents, the committees of each village, and the municipality would have different interests in development directions for local land use and rural development planning. In this context, the idea of enabling local populations to present their common interests during planning and decision-making has therefore received increasing attention from Chinese authorities.

Given this history, our study analyzes how the involvement of public citizens in a Chinese context may enrich spatial planning with respect to ideas and solutions. We accomplish this by selecting a recently approved landscape corridor plan, investigating how alternative plan solutions could have been developed through local resident involvement, and further reviewing these alternatives with a group of professional planners who were involved in the original planning process.

## 4. Material and Methods

### 4.1. Study Area and Current Corridor Plan

A Silk Roads cultural landscape corridor plan in a suburb area of Zhangye Municipality (Figure 1) was used to analyze a local, recently approved landscape corridor plan, and to investigate the implications of public participation for innovating planning content. The corridor plan was first proposed by the local municipality as a part of a conservation project along the Silk Roads region in Western China in 2015. Since the Silk Roads were added onto the list of World Heritage Sites in 2014, the municipalities located in these regions began to promote various local cultural landscape conservation and development projects. There exists as well an overall regional plan that aims to protect local natural resources and heritage sites, improve the local agricultural food production industry system, and promote regional cultural and ecological tourism development. The regional development of the whole Silk Roads cultural landscape corridor will be guided by a number of local corridor plans. The case analyzed in this paper is one such local plan.

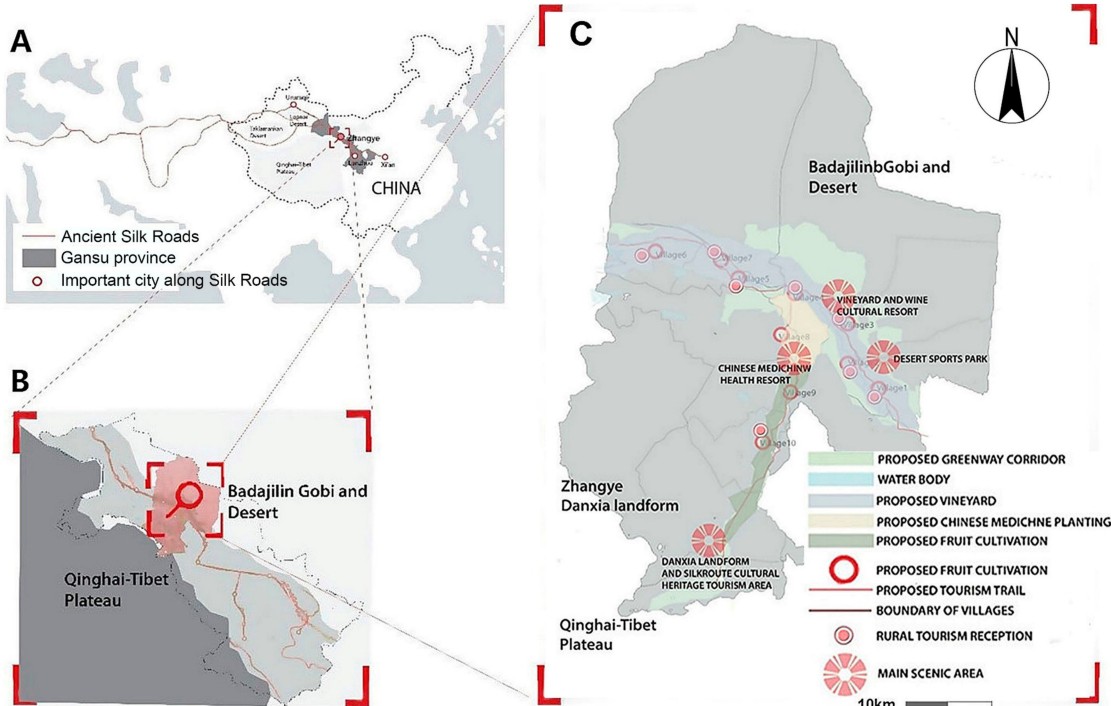

**Figure 1.** The location of the Zhangye cultural landscape corridor plan along the Silk Roads in Northwestern China. (**A**). General map of the Silk Roads area; (**B**). Our study area within the Silk Roads region, surrounded by the Badajilin Desert in the north and the Qinghai-Tibet Plateau in the South; (**C**). The current Zhangye cultural landscape corridor plan, including the proposed tourism trail and proposed development package projects (approved by local decision-makers).

Our study area is located in the narrow central part of the Heixi Corridor, historically known as the sole route used by ancient Silk Roads traders to pass between the northern Badajilin Desert and the Qinghai-Tibet Plateau in the South (Figure 1A,B). The existing Silk Roads linear heritage within Zhangye Municipality includes stretches of the Great Wall of China along with numerous temples, heritage trading sites, cave sculptures and other architectural remnants of historical civilizations and their early cultural exchanges with Europe.

In 2015, Zhangye municipality's corridor project started with the idea of building a recreational trail that would revitalize the region by connecting these Silk Roads cultural sites. Prior to plan implementation, pedestrian and tourist access to these cultural sites, or even to the scenic landscapes between them, was characterized by discontinuity. Hence, decision-makers from local municipalities began looking for a way to connect existing Silk Roads cultural resources while also enhancing opportunities for recreational and tourist activities via a connected corridor plan. Developed by China Agricultural University and with the support of local planning departments in 2017, one such corridor plan was then adopted by Zhangye municipality. The aim of that plan was to preserve and strengthen the existing landscape patterns and vegetation, while at the same time emphasizing the aforementioned historical and cultural areas. In short, the plan would combine the local agricultural and ecotourism industries for the betterment of regional sustainable development. With a positive outlook, this proposal for the cultural landscape corridor was approved. Implementation spanning the area between ten different villages began in Summer 2018. The current cultural landscape plan, presented in Figure 1C, includes: (1) a greenway corridor that contains a newly planted vineyard along the river and a recreational trail, as well as plans for a wine cultural resort; (2) a desert sports park; (3) a Chinese medicine garden; (4) a Chinese medicine-focused health and culture resort; (5) a rural recreation system with access to the river tributary, including access to eight tourism service centers, reception centers, and homestay dwellings along the recreation trail; (6) a fruit cultivation area; and (7)

a central Danxia landform and Silk Roads cultural tourist area. The tourist area will be built in the pseudo-classic style of a tourism village, with the Silk Roads cultural features being the focus of the outside reception areas and the Danxia landform scenic area visible to the West.

*4.2. Methods*

Based on our analysis of the current plan, a two-step framework was designed to study the potential enrichments that could be gained by including local resident participation in building the final plan: (1) Exploring the conflict area through different development preferences between local residents and decision-makers; and (2) Assessing alternative future development through scenario planning. Our final study applied a variety of different methods and research techniques, including local resident participatory mapping surveys, spatial analyses (Kernel Density analysis), semi-structured interviews, group meetings and workshops, and scenario planning.

4.2.1. Conflict Areas with Different Development Preferences Between Local Stakeholders and Decision Makers

Different stakeholders—from farmers, to local tourist operators, village residents, political decision makers, or professional planners—are likely to have different perspectives and preferences concerning spatial development [19]. Conflicts may therefore appear during the planning process, specifically concerning the promotion and development of different land use areas. In this study, we explore such conflict areas by comparing elements of the current plan with the spatial implications we create based on local stakeholder preferences.

4.2.2. Building the Map of Local Resident Preferences

The spatial patterns that arose from an analysis of local resident views were built via a participatory mapping practice using local residents' development preferences. We performed this participatory mapping survey in January 2018, when the research team, with the help of the municipality's planning department, was introduced to the committees of ten villages along the Silk Roads corridor. Our researchers went to these 10 villages and informed local residents about our roles, the surveys, and our main aims thanks to the committees in each village. We then selected our participants randomly and independently from this committee sample. Finally, 20 participants from each village were invited by researchers, through snowball sampling (*n* = 200), to participate in small group interviews. We began these interviews by showing each two- or three-member group a satellite image of the village in question and a paper map covering the ten villages, thereby displaying current land cover, as well as the locations of current residential settlements. These steps were used to help respondents identify the locations of their familiar landmarks, as well as the location of each village within our study area.

Next, we used the corridor planning proposal to explain future area plans to participant-residents. Then, in response to our questions, these individuals were asked to mark which areas they thought were the most suitable toward the future planning of each kind of development preference, using questions like, "Which place do you think is most suitable for planning a desert sports park within this area?." Respondents' development preferences were then mapped using a pencil to draw points on the ten-village paper map mentioned above. To gain a deeper understanding of these preferences, our researchers then asked participants about the reasoning behind their chosen preferences.

From there, we digitalized the mapped points into ArcGIS (version 10.6, ESRI, Redlands, CA, USA) and used a Kernel Density analysis that together produced an accurate spatial distribution of our gathered data. We then applied the Kernel Density analysis again, this time performed by matching each point with its nearest neighbour using hierarchical clustering. Our active variables became those "hotspots" on our spatial pattern showing these participant data-points, revealing which areas attracted the most attention on our Kernel Density heat maps. Using this heat map, our team was thus able to gather a quantitative overview of the development preferences of our local participants.

4.2.3. Overlapping the Spatial Patterns of Local Resident Views with the Current Plan

The current corridor plan also contains a spatial outline that indicates the objectives and development directions of the Silk Roads area based on decision-makers' preferences. By overlapping the spatial patterns we had documented from local resident preferences with those of the current corridor plan, we could discover and further explore potential conflict areas between the two sets of development preferences. The layout of development projects on the current plan (e.g., vineyard corridor, desert sports park, rural tourism greenbelt) reveal the development preferences and patterns of the decision-makers involved in initial corridor planning. As such, overlapping each group's preferences also helped us analyze which areas had similar data stratification, showing us where the development preferences of each party are compatible, or similar. From there, we selected those villages where local residents had few or no development preferences, indicating the most profound conflict areas are between the two groups.

4.2.4. Alternative Futures Assessment Through Scenario Planning for the Conflict Area

Scenario planning is a method used for making long-term plans about the future, with the aim of fair decision and policymaking. This practice is widely used in geography, business, and politics [30] where scenarios are identified as a description of the possible future state of something, including event sequences that could lead from the current state of affairs to (or toward) a preferred future state [31]. We completed several alternative futures assessments through scenario planning for the conflict areas as mentioned in the above overlapping step. Based on the different development preferences of local residents and actual decision-maker, we constructed different scenarios to describe plausible alternative futures of our focus corridor planning area. While there is no single approach to scenario planning, and related literature shows various methodologies for building such scenarios [34,35], scenario planning generally emphasizes the following steps:

(1)     Identify and establish the driving factors of change;
(2)     Facilitate scenario-building and visualization;
(3)     Elicit issues and qualify the impact of each scenario [36]

In this context, the existing corridor plan—with updated development directions—would be considered to represent the driving force behind changes to the local land. As a consequence, alternative futures based on either resident or decision-maker preferences were used as our scenario guidelines. It turned out that photorealistic visualizations based on land photos comprised an efficient tool for presenting planning metrics [37] (less-effective tools included GIS-based modelled landform surfaces, drawing, figures, etc.) [38,39].

Further exploration of scenario-planning itself showed that this method is especially useful for visualizing smaller-scale projects because it enables the presentation of details that are important for non-expert stakeholders [36,40]. With these considerations in mind, our team selected photorealistic visualizations as our primary tool for explaining and contextualizing our scenario discussions with non-expert, resident stakeholders. Following this visualization process, we then discussed our findings with these stakeholders during group meetings, using these mapped tools to best present the planning, construction, and function issues of each scenario, as well as their impact on local development.

Our study used a single, bird's eye view photograph rendered using Google Earth's satellite imagery, adding to it our data relating to those villages with development planning conflicts between local residents and decision-makers. Using Photoshop CS 6.0 (Adobe, San Jose, CA, USA), we further visualized our given scenarios by adding different layers, each containing different elements of our alternative future assessments to the original bird's-eye photograph.

During the process producing these photographs, significant new landscape elements of each alternative future were turned "on" or "off" depending on responses from the stakeholders' development preferences. Gradually these photographs were supplemented with detailed illustrations describing the full spectrum of activities and infrastructures each scenario represented. All of these

materials helped the stakeholders to better understand the areas. We then used the two bird's eye photographs to discuss the scenario planning with different stakeholder groups, which includes three steps:

(1)   Verification;
(2)   Combination;
(3)   Assessment.

First of all, we performed a verification process: each of the two bird's eye photographs presented an alternative future based on separate expressions of the two sets of development preferences (residents and decision-makers; scenarios 1 and 2, respectively). The alternative futures visualized in two additional scenarios were subsequently discussed with decision-makers as well as small groups of residents. Based on feedback from resident groups, we modified the then-current scenario 1 into one that was even more consistent with the residents' development preference. We made a new scenario 2 by performing the same process with decision-makers (Figure 2). All of this is a part of verifying that our constructed scenarios corresponded to the preferences of the two groups.

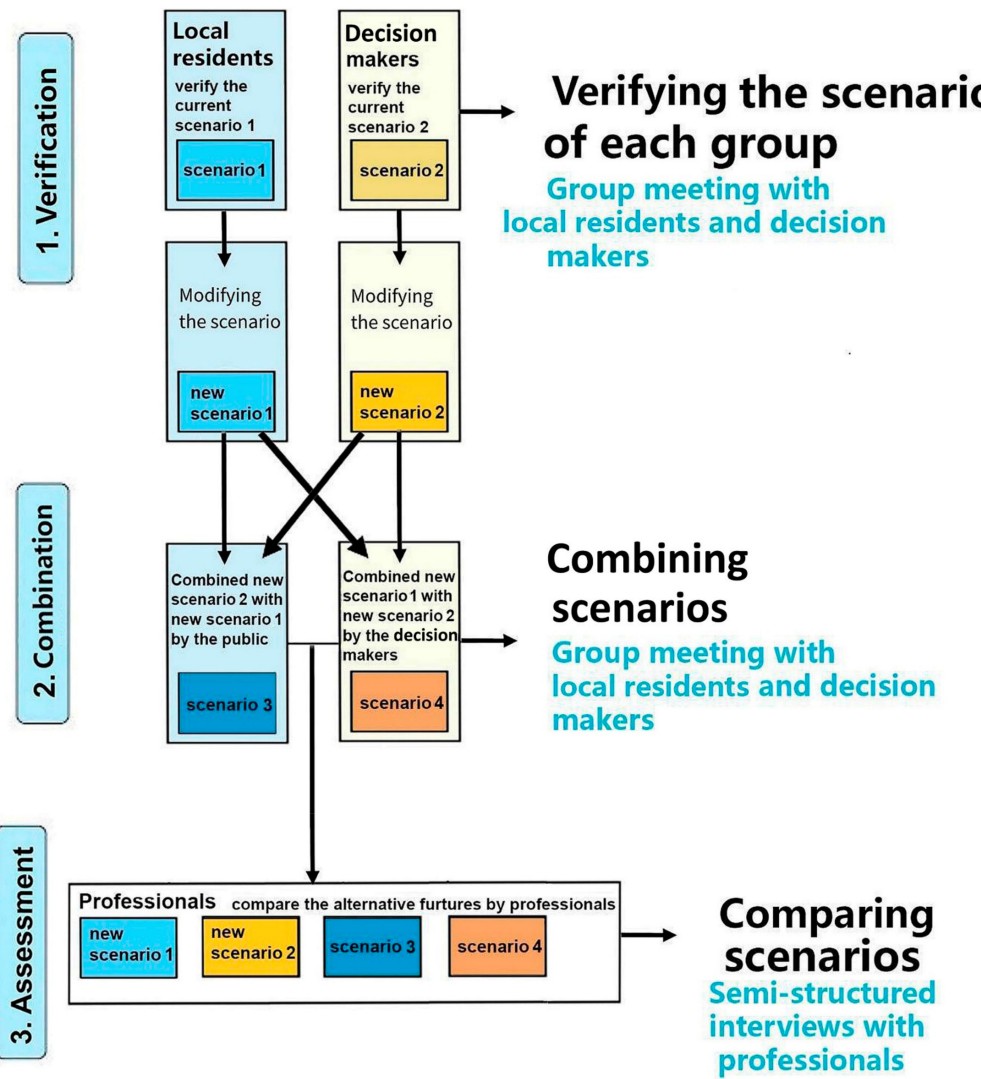

**Figure 2.** Framework of alternative futures assessment through scenario planning.

For step two, respondents were asked to discuss the differences that arose when combining the development preferences of both residents and decision-makers. We then modified and improved the

new scenarios 1 and 2 to become scenarios 3 and 4 by combining these two different group opinions (Figure 2).

For step three, we invited five professionals who participated in the original Silk Roads corridor project to examine our four scenarios, as well as share their views as to the impact of each scenario. These conversations took the form of semi-structured interviews. The five professionals had already worked as a team together for one month on this project. All of them participated during the whole process, from fieldwork to final overall proposal (Figure 1C). However, they were not aware of details within the proposal, or specific impacts on specific villages. In this respect, they could be regarded as the outsiders (third parties) who were familiar enough to the case study are to assess the scenarios. We sent the photos of these scenarios to them one day before the interview and invited them for individual assessment interviews. Each interview lasted from 30–60 min and concerned how the individual professional valued each scenario, including how he or she ranked them in order of suitability degree for local development, and the reasons behind the ranking.

## 5. Results

### *5.1. Areas With Conflicting Development Preferences*

#### 5.1.1. Characteristics of Respondents

Our survey covered full- or part-time local residents from ten villages. In total, 200 respondents participated in the survey (53% male and 47% female). Sixty-nine percent were farmers, with the remaining residents being either technology staff (11%), administrators (6%), tourism service (5%), students (9%) or jobless (1%). Thirty-eight percent of respondents had been to high-school and 32 percent were under 30 years old, while 47 percent were between 30 and 60, and two percent were above 60 years old. The majority of our respondents had been settled in the local area of study for more than 20 years (73%).

The questions we asked directly concerned which land use developments local residents preferred. The most frequently supported development preferences were rural tourism and homestays (27%), followed by fruit cultivation (23%). The desert sports park idea (6%) is less attractive for local residents when compared with other development opportunities.

#### 5.1.2. Spatial Patterns of Development Preferences

A Kernel density analysis of the development preferences of specific sites based on local resident perspectives is shown in Figure 3. Similar preferences for rural tourism and homestays were present across all villages within the corridor proposal coverage. Our heat map shows that the highest degree of clustering for given development preferences pertaining to the vineyard and wine cultural resort were common to the southeast part of the region (village 1, village 2). Support for the Danxia landform and Silk Roads cultural heritage tourist sites also clustered in these villages, along with village 3. For the Chinese medicine health resort and the desert sports park, support was relatively limited, and present only in villages 8 and 9.

A summary of the compatibility between the development preferences of local residents and decision-makers is as follows: Concerning rural tourism greenbelt planning, no major conflicts were found in nine of the ten included villages (1–9). Decision-makers and local residents shared a common preference for building a Chinese medicine-focused health resort and medicinal garden in villages 8 and 9, and showed a similar interest in developing a vineyard corridor and wine cultural resort in villages 2, 3, and 4. Based on the consensus of local residents and decision-makers, villages 8 and 9 were also considered as potential areas for the cultivation of fruit.

At the same time, there were a few significant differences between the development preferences of decision-makers and local residents. Most importantly, we found that residents frequently pointed out village 1 as the best area for developing a Danxia Landform and Silk Roads cultural tourism area,

as it already had red cliff landforms and some Great Wall historical importance. At the same time, decision-makers in the real plan had considered a bare land south of village 10, because of its existing National Danxia Landform Geopark along the Silk Roads corridor.

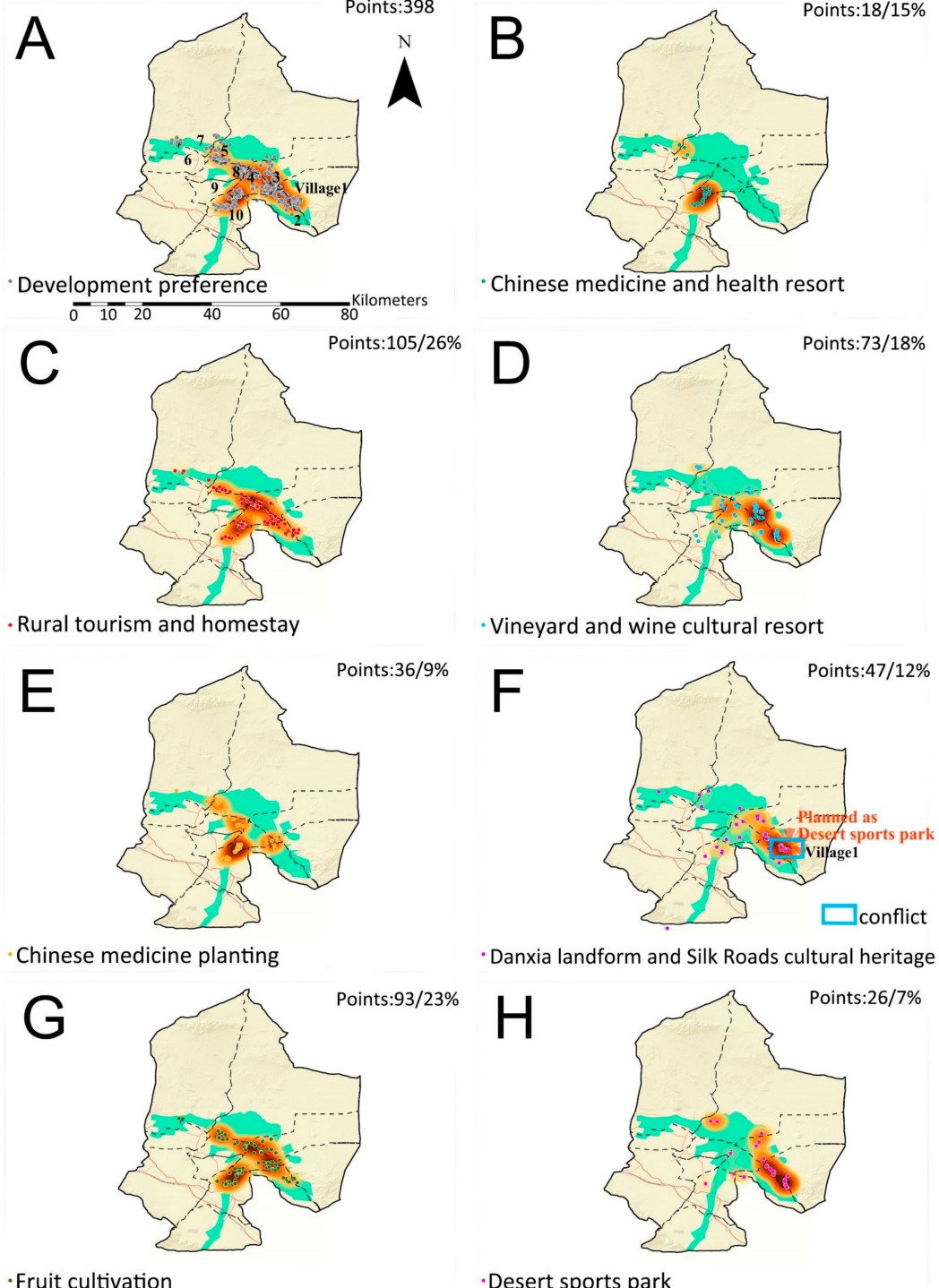

**Figure 3.** Kernel density heat maps of development preferences for local residents. (**A**) All kinds of development preferences; (**B**) Chinese medicine health resort; (**C**) Rural tourism; (**D**) Vineyard and wine cultural resort; (**E**) Chinese medicine planting; (**F**) Danxia landform and Silk Roads cultural tourism area (as the location of the planned desert sports park); (**G**) Fruit cultivation; and (**H**) Desert sports park. A higher density of points is visualized in dark purple, with lower densities in light purple.

To summarize, we narrowed down the major conflict areas to be within the development preferences of village 1—which included differing local resident preferences for the Danxia landform and Silk Roads cultural tourism areas, and different decision-maker preferences for the Desert sports park—and of village 5, which included differences in local preferences for fruit cultivation as well as in decision-makers' preferences limiting the development to a vineyard corridor. In this case, there are no 'hotspots' of development preferences for local residents in village 10, even though decision-makers have in fact planned the Danxia landform and Silk Roads cultural tourism areas there.

When we started this research, both the municipality and all local village committees had confirmed the proposed village 1 desert park. Meanwhile, village 5 merged with another village and a new village committee had been elected. To date, the vineyard corridor proposal has not yet been confirmed at the village level, and the new committee is considering changes. On this background we selected village 1, which has an unambiguous conflict between the preferences of local residents and current proposal (Figure 3F), for further analysis and assessment. Our goal was to work toward a better and more focused understanding of possible alternatives (or combinations of alternatives) during the next phase of our study.

### 5.2. Comparing the Differences between Alternative Futures through Scenario Planning

Within our assigned conflict area, we then made our detail bird's eye view photo of the planned sample area for scenario planning in village1, following different development guidelines (Figure 4A–C). The local original landscape includes cultivated land along the river, artificial surfaces, and bare land with red Danxia landforms (Figure 4B,C). The guidelines for scenario planning included (1) Local resident development preferences: Danxia landform and Silk Roads cultural tourism areas, and (2) Decision-makers' development preferences (the current plan): Desert sports park.

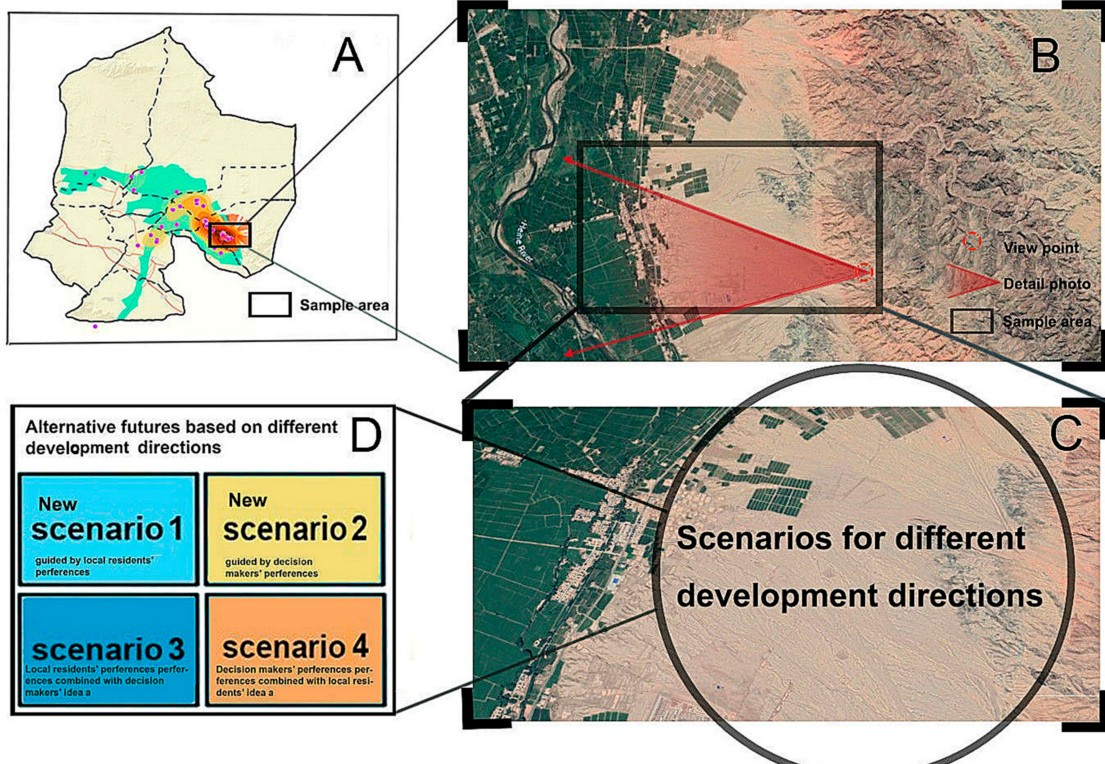

**Figure 4.** The four conflicting scenarios for village 1. (**A**) Conflict area in village 1; (**B**) Detailed view of the project area: all circular images in subsequent figures are visualizations of this original landscape; (**C**) The Enlarged area for scenario planning (**D**) Four scenarios based on different development directions.

After a thorough discussion with local residents and decision-makers during our scenario verification and combination process, the following four sections with subtitles describe the scenarios that we finalized as Figure 4. The scenario descriptions are followed by the results we collected based on local resident and decision-maker comments during group meetings. In addition, these scenarios reveal the presence of the planned sample area in relation to the current plan, in the event that one of the four development directions dominates after the Silk Roads cultural landscape corridor plan has been completed (Figure 4).

Finally, we created four scenarios of the future, each guided by a different set of development preferences (see Figures 5–8). For better presenting the differences of different scenarios; we enlarged the landscape change area as Figure 4C. These assumptions are of crucial importance toward an accurate presentation of the complex processes and alternative futures that shape landscapes in general, and this narrow corridor in the specific.

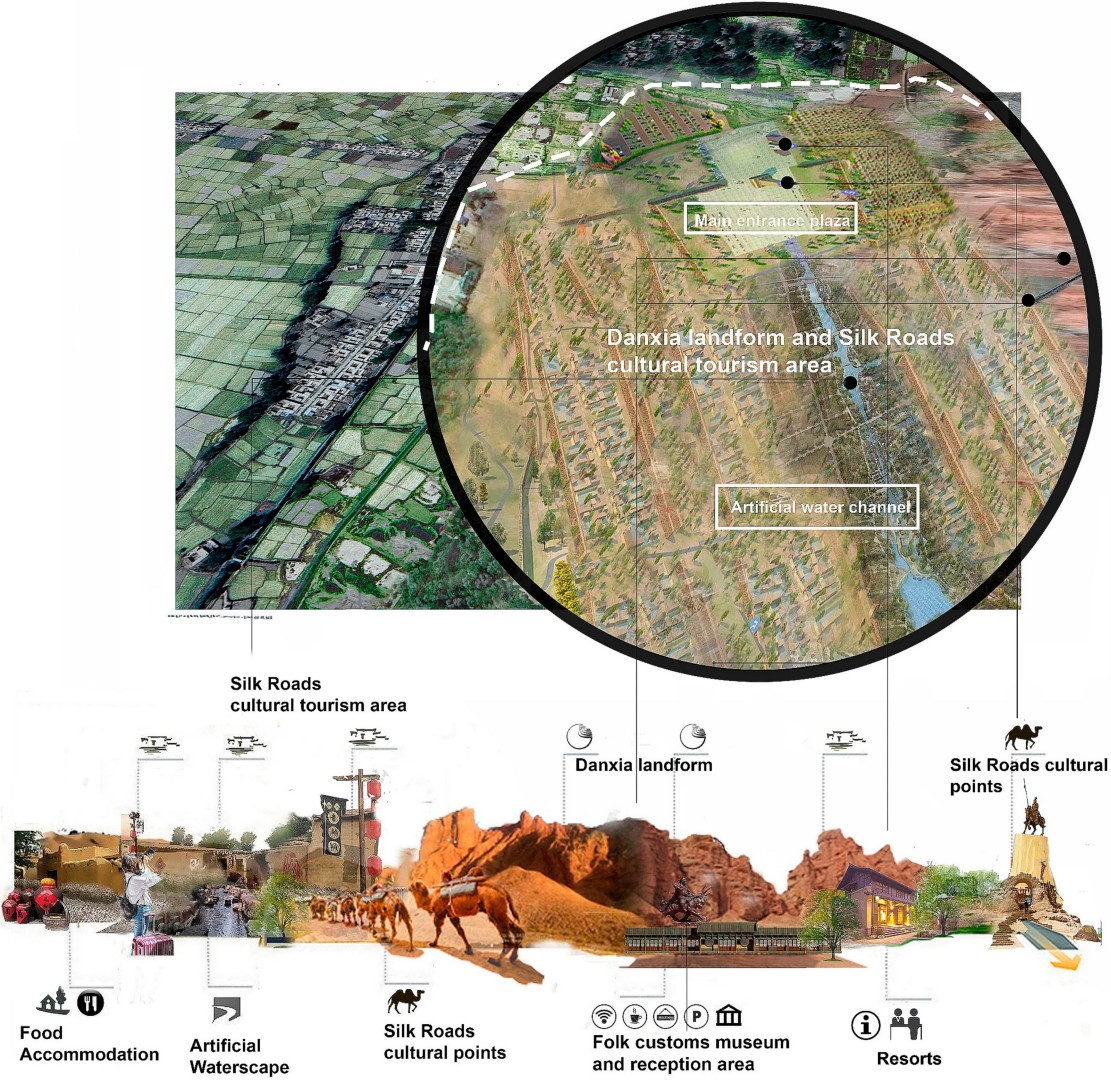

**Figure 5.** Scenario 1, guided by local resident development preferences. (The figure is a detail from the planning area of the scenario 1 poster shown to the local residents.).

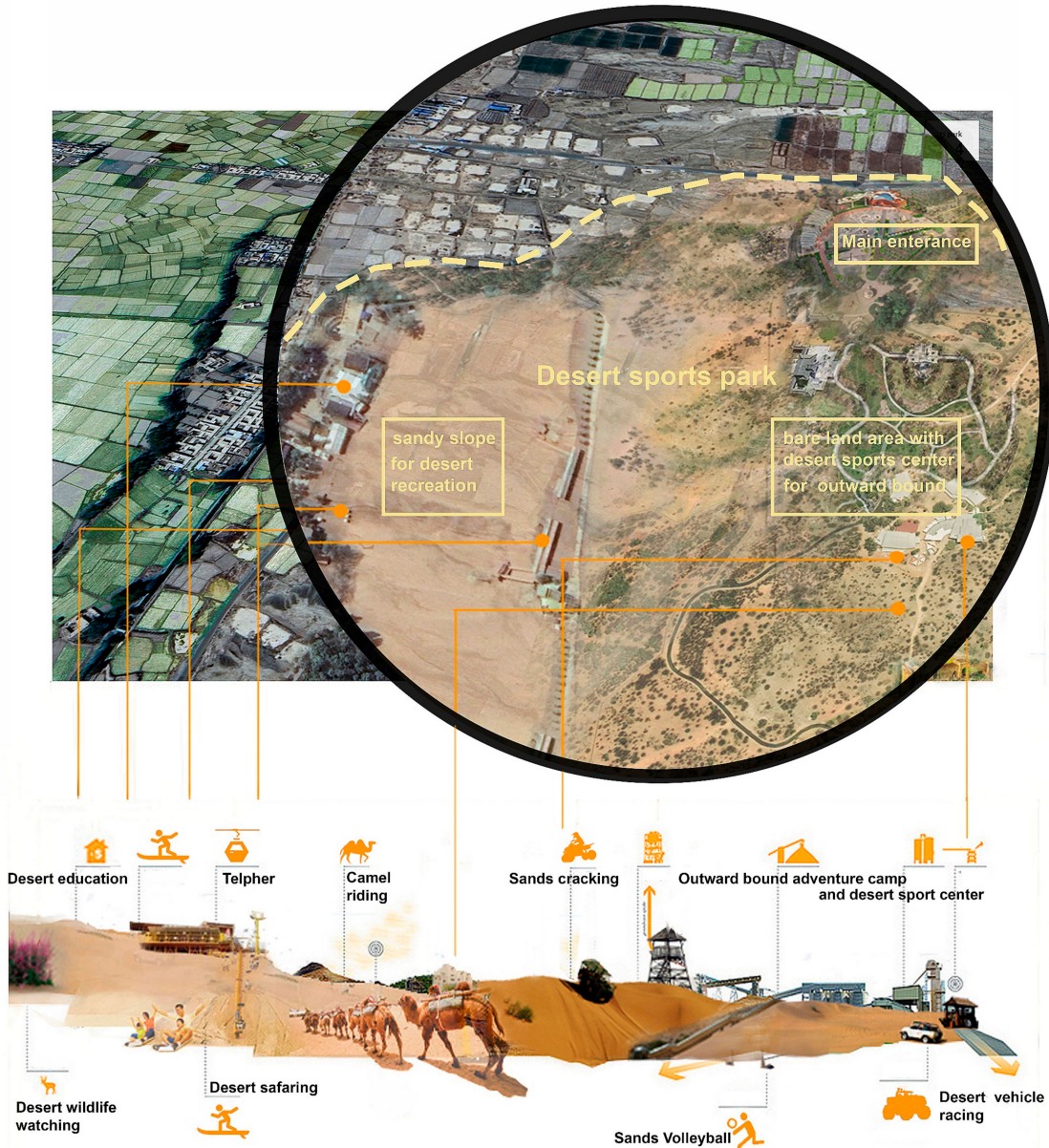

**Figure 6.** Scenario 2, guided by decision-maker development preferences based on the current plan (The figure enlarged the planning area of the poster of scenario 2 shown to the decision-makers).

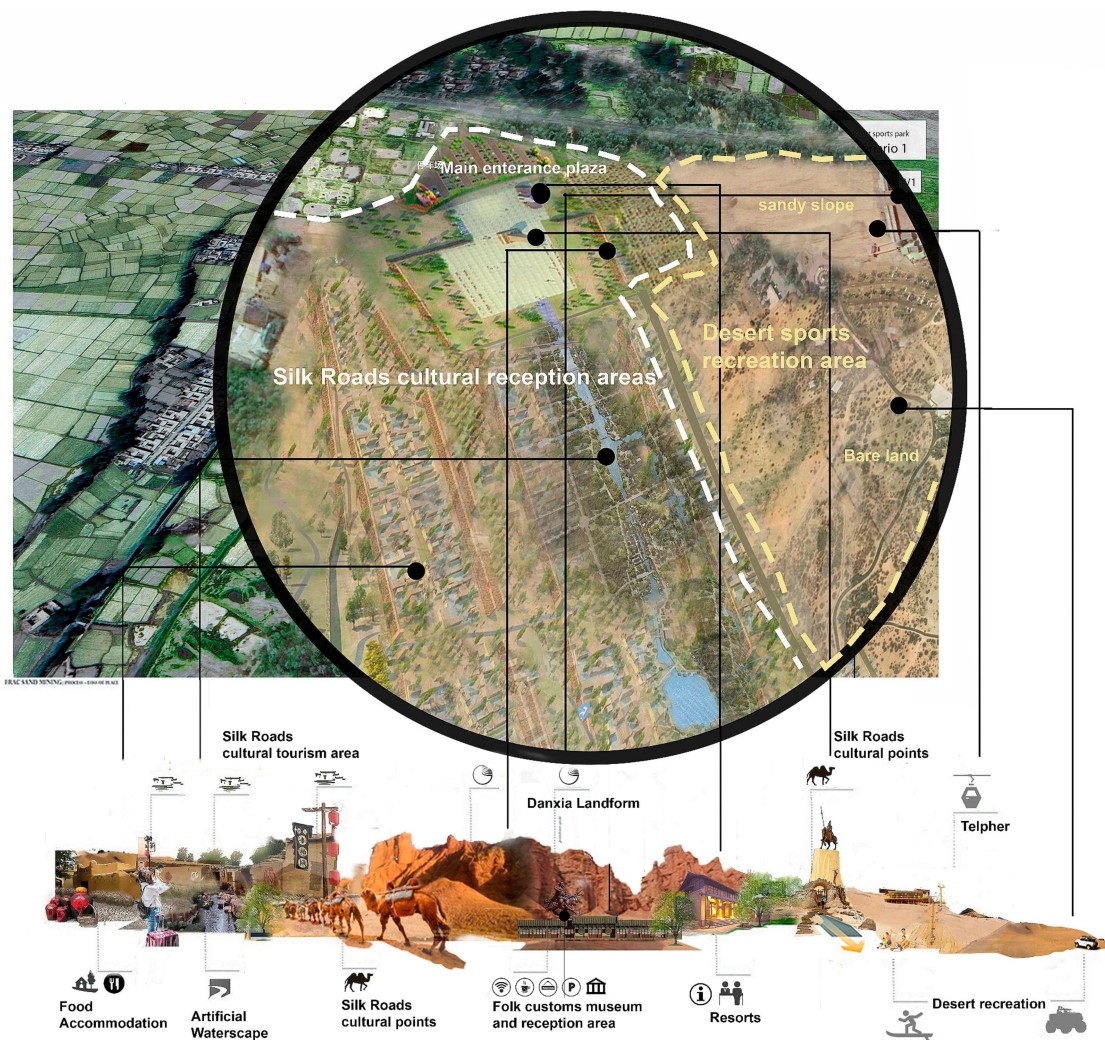

**Figure 7.** Scenario 3, guided by local resident development preferences combined with decision maker ideas (The figure enlarged the planning area of the poster of scenario 3 shown to the local residents).

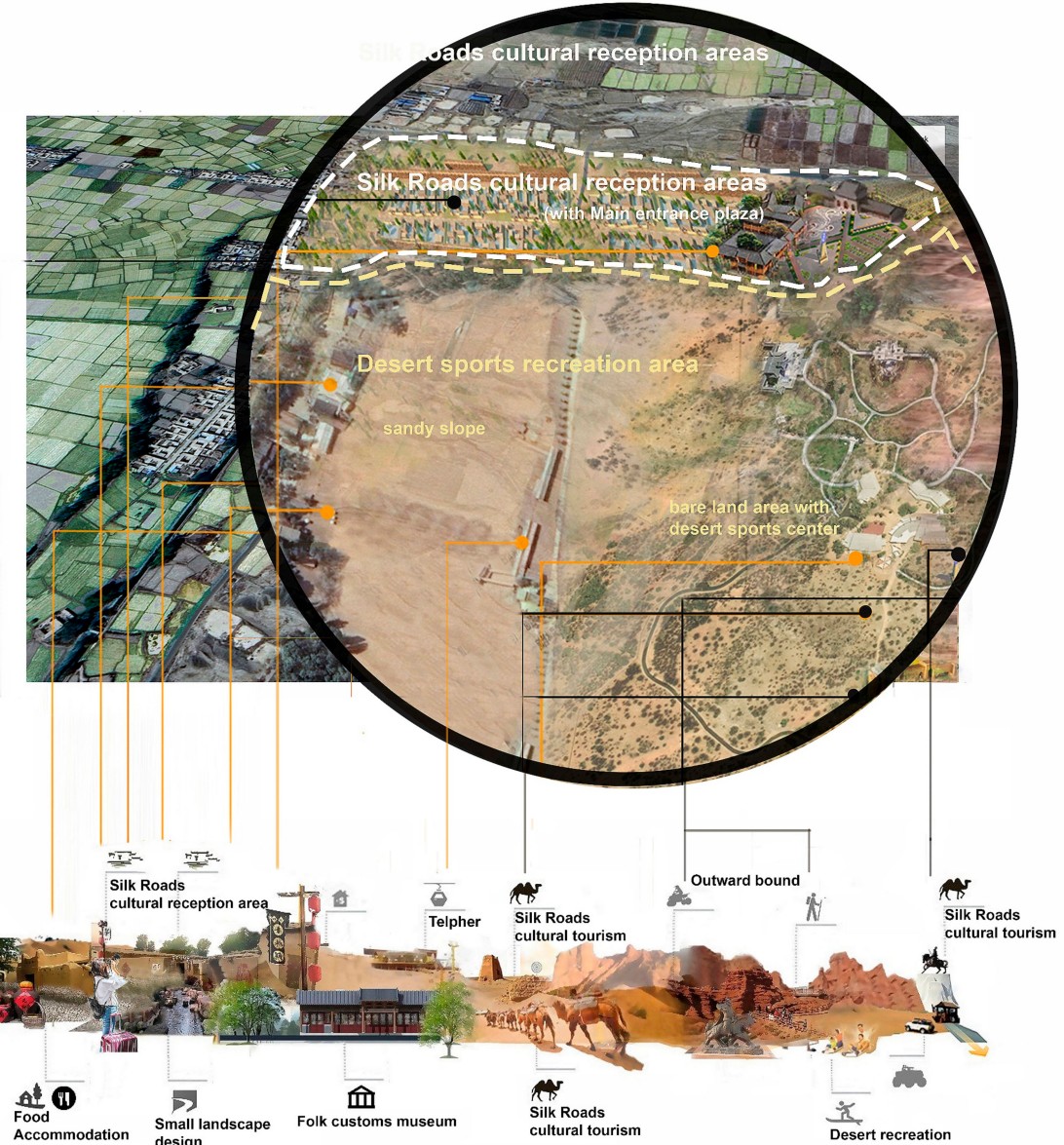

**Figure 8.** Scenario 4, guided by decision-maker development preferences combined with local resident ideas (The figure enlarged the planning area of the poster of scenario 2 shown to the decision-makers).

### 5.2.1. Scenario 1—A Scenario Guided by Local Residents' Development Preferences

In our first scenario, we assume that the area has been developed based exclusively on the preferences of local residents, meaning it has become a tourist area focusing on the Danxia landform and Silk Roads cultural histories. The plan mainly consists of a large central plaza, a recreational and residential district built in the historical style, and a temple with adequate open space. The entrance plaza includes a folk customs museum, a sculpture of a local legend and hero, and other supporting amenities like parking areas. The folk recreational and residential district includes various recreational and reception facilities such as shopping streets, homestay venues, and distinctive small hotels, restaurants, and bars; the recreational and residential districts are separated by a landscaped artificial water channel. At this future moment, planning continues for a Loong Temple at the end of the channel, complete with open space for local market days and religious festivals. In this scenario, this cultural landscape corridor project provides cultural education, recreation and tourism functions relating to the Silk Roads culture, experiences, and folk customs, all with local Danxia landform scenery as the background.

### 5.2.2. Scenario 2—The Scenario of Decision-Makers' Development Preferences

In the second scenario, which visualizes decision-makers' development preferences as reflected in the current plan, we see an area that has developed as a desert sports park attractive to tourists. According to the requirements outlined by decision-makers, the desert sports park consists of a main gate (with supporting facilities such as a parking area), a large-scale sandy slope for desert recreation, a large-scale bare land area as the desert sports center, an outward-bound camp, and a small garden to commemorate the ruins of a Silk Roads fire tower. The sandy slope area contains facilities for sandboarding, desert safaris, sand cracking, camel riding and a telpher. The bare land area includes hiking trails that facilitate the exploration of the Danxia landforms (with the roads and spaces also built for cross-country training and other events like mountain biking and off-road vehicle racing), a reception centre for desert sporting events, and an outward-bound camping area for outward-bound and desert adventurers. In this scenario, the desert park has been established as a unique tourist attraction along the regional cultural landscape corridor, and the developed area provides all the functions expected for cultural heritage conservation, desert recreation, hosting sports training and events, and professional outward-bound activities.

### 5.2.3. Scenario 3—A Scenario Guided By Local Residents' Development Preferences Combined with Decision-Makers' Ideas

In our third scenario, local resident development preferences are combined with existing decision-maker ideas, and we visualize a Danxia landform and Silk Roads cultural tourism area, complete with desert recreation facilities. The layout of the large center plaza, recreational and residential district is built in the historical style, and includes a temple with open space. For this alternative future, facilities have been added to the hillside for desert recreation, such as sandboarding on the sandy slopes, and a telpher connects tourist populations to the Danxia landform. In addition to the main function of cultural education, this scenario focuses on Silk Roads cultural recreation experiences and folk custom tourism, while also providing some desert recreation amenities for more adventurous travellers.

### 5.2.4. Scenario 4: A Scenario Guided By Decision Makers' Development Preferences Combined with Local Resident Ideas

In the fourth scenario, we assume that identified conflict areas have been developed according to decision-makers' development preferences paired with ideas from local residents. The area has therefore become a desert park with Silk Roads cultural reception areas near the main entrance. This scenario consists of a Silk Roads cultural reception area, an entrance plaza, a large-scale sandy slope area for desert recreation, a large-scale bare land area with an accompanying desert sports centre, an outward-bound camp, and a small garden to commemorate the ruins of a Silk Roads fire tower. The sandy slope and bare land areas provide similar functions as desert recreation amenities, as well as supporting facilities for desert hiking trails and outward-bound centres. The entrance plaza includes a folk customs museum and a main gate as historical features, while the reception area includes specialty restaurants and homestays that are located close to existing villages. In this scenario, this area acts as a functional source for cultural heritage conservation, cultural and folk experience-based tourism, desert recreation, sports training and recreation, and professional outward-bound activities.

### 5.3. Professionals' Responses to Possible Future Scenarios with and without Local Resident Participation

After scenario planning, the five professionals involved in the current corridor plan proposal included in our study were interviewed to provide an additional professional assessment of the four scenarios. The five professionals ranged ranging in age from 26 to 41, and two were female while three were male. Each professional came from a different (but relevant) educational background (geography, landscape architecture, urban planning, and architecture), and thus played a different role in developing the current corridor plan.

### 5.3.1. Differences between Scenarios

When the professionals compared each of the four scenarios, they discovered that any differences between them could be generalized as going in two main directions: on the one hand, scenarios 1 and 3 represented a direction toward cultural tourism, whereas scenarios 2 and 4 focused more on a recreational desert sports park. As a result, the professionals found that scenario 1 was the only of the four to have cultural characteristics mainly consisting of historical buildings. By contrast, these professionals felt scenario 3 included more desert-oriented recreation facilities than scenario 1, owing to the presence of the small-scale desert recreation area planned for that scenario.

Scenario 2, on the other hand, focused on the building of a desert sports park that would contain professional recreation facilities. The reception areas in scenario 2 were planned mainly for cross-country training and camping functions. Out of the four, scenario 4 shows a desert park that integrates characteristics of the Silk Roads culture into supporting facilities that can then be used for the leisurely reception of regular tourists. The professionals also noted differences between scenarios 3 and 4, where each combined the ideas of local residents and related decision-makers:

*"When compared with scenario 3, scenario 4 did not include many of the 'unreasonable' design elements supported by residents. A large-scale shopping street feature, or a canal for water scenes would, for instance, be very difficult to maintain, and would affect the local environment. As these are both 'unreasonable' design elements for this area, scenario 4 instead developed an industrial output centered on the main desert resource."* (Professional 2)

*"Scenario 4 has better functional zoning than scenario 3. The functions in scenario 4 have a clearer focus on desert recreation, complemented by the supporting facilities containing local features, while the functional zoning in scenario 3 looks more piecemeal."* (Professional 1)

### 5.3.2. The Most Suitable Scenario for Local Sustainable Development

The professionals were asked to rank the assessed potential alternative futures for local development in the four scenarios, as shown in Table 1 below. Scenario 4, guided by decision-maker development preferences, and combined with the desires and ideas stemming from local residents, was evaluated as the most positive alternative future for local development.

**Table 1.** Outcome of rankings and selection of the most suitable scenario.

| Interviewee | Most Suitable Scenario | Scenario of Second Rank | Scenario of Third Rank | Scenario of Fourth Rank |
| --- | --- | --- | --- | --- |
| Professional 1 | Scenario 4 | Scenario 2 | Scenario 3 | Scenario 1 |
| Professional 2 | Scenario 2 | Scenario 4 | Scenario 3 | Scenario 1 |
| Professional 3 | Scenario 4 | Scenario 3 | Scenario 1 | Scenario 2 |
| Professional 4 | Scenario 4 | Scenario 1 | Scenario 2 | Scenario 3 |
| Professional 5 | Scenario 4 | Scenario 2 | Scenario 3 | Scenario 1 |

Four of the five listed professionals found scenario 4 to be the most suitable development scenario, while professional 2 preferred scenario 2 and listed scenario 4 as the second-most suitable, in their opinion. From our report, these professionals also thought scenario 4 has the potential to combine local cultural features, folklore characteristics, and desert recreation, all unified by characteristic tourism features that support the main service areas and ultimately enrich the local tourism production chain. It should be noted that this scenario makes full use of local natural resources within a reasonable investment. As one of the professionals explained during the interview, for example:

*"Scenario 4 has the most diverse facilities for regional development. I really like the idea of combining desert sports recreation with local Silk Roads cultural elements for the supporting facilities. For example, I like the idea of involving the 'Journey to the West' myth in the design of local resident*

*spaces as well as trails in the desert sports park. This scenario also highlighted the natural features of the desert and Danxia landforms. It could become a unique attraction in this corridor region among other places referred to by the current proposal. Compared with scenario 1 and 3, scenario 4 demands less investment. And I also like that its functional zoning design fully used local resources."* (Professional 3)

Another professional found that:

*"Scenario 4 is the most suitable scenario for me. Firstly, it has the most fully developed functionality for both the desert and the local Danxia landscape features. The desert recreation elements could create a unique tourist attraction for the benefit of the corridor region as a whole. In addition, the desert sports area could attract a strong, long-term source of visitors attending sports events such as hiking or cross-country vehicle racing. Moreover, there is a tourism-supporting service area that bears local cultural characteristics, combined with the desert recreation and folklore experience in this area. Tourists might stay longer where the amenities are best, which inflates local consumption growth and thus benefits residents. Last but not the least, this scenario has the most comprehensive functionality for its related economic investment. Compared with scenario 1 and 3, this scenario requires less construction work volume, while also fully covering the recreation amenities for both the cultural tourism and desert sports features at the same time. It also makes full use of local natural resources, for example the desert sports area in the desert landscape and the hiking trails located in the Danxia landform area. This scenario minimizes environmental consequences due to its more economical plan for construction work volume".* (Professional 5)

One professional regarded scenario 2—"guided by decision-maker preferences in the current proposal"—as the most suitable direction for development. He states:

*A desert park with a desert recreation theme as a priority service function could highlight the particular organic qualities of this area. This area has special natural resources, as well as Silk Roads cultural resources that can be found along the whole corridor region. This scenario has an intense purpose.* (Professional 2)

Nevertheless, this professional evaluated scenario 4 positively:

*"Scenario 4 requires the smallest construction work volume when compared with other scenarios, and provides space for potential changes in planning, all at the lowest cost."*

5.3.3. The Scenarios with Negative Comments for Local Development

Overall, none of our professionals found scenario 1 (guided by local resident development preferences) or scenario 3 (guided by local resident development preferences combined with decision-maker ideas) to be very engaging. Each of the professionals mentioned that the high construction investment involved—not to mention the problems of undesirable competition between this proposed Silk Roads tourism town and other villages along the corridor—was the main reason that they ranked these scenarios poorly. As shown from the testimonies below, most felt that these scenarios are not realistic for creating sustainable development. One professional pointed out that a large initial investment in these historical buildings is not wise, as it could decrease much-needed room for future adjustments and thereby affect the flexibility the project has for future improvements.

*"The scenario focusing on the Silk Roads cultural tourism area looks tedious. The scenario that combines elements of the Silk Roads cultural tourism area with some of the desert recreation facilities looks better; however, people would not be able to effectively use these facilities as the desert recreation area is quite small compared with other scenarios. In addition, due to the large scale of historical style buildings such as the folklore expo museum and the shopping and recreation streets, there is limited space for future adjustments if new projects demand development in this area.* (Professional 1)

Another stated that

> *"The scenarios involving the Danxia landform and the Silk Roads cultural tourism area require a large amount of investment for building each block in a historical style. And there will be a Danxia landform-Silk Roads characteristic tourism area planned along the peripheral zones of the national geopark area of the Danxia landform in the current corridor plan anyway. The focus of developing a cultural tourism area in this village has no advantages in relation to the homogenous competition."* (Professional 2)

A third professional summarized his view of scenarios 1 and 3 in this way:

> *"These scenarios will tend toward development in the direction of commercialization, and is likely because of other tourism-estate projects. There are already similar projects along the Silk Roads. Thus, these developments would not highlight the local features of the area. Besides, the 'cultural tourism area developing direction' requires more investment for building and maintaining properties and attractions, which places a high risk on the future of the whole operation."* (Professional 5)

The interviews showed that four of the five professionals recognize that scenarios with different degrees of local resident involvement indeed facilitated new and innovative thinking into each applicable area's regional development when compared with the scenario 2 based on the current plan from important decision-makers alone. Each scenario that included the opinions and requests of local residents often brought in new and creative ideas not considered by current decision-maker plans—such as expanding the tourism market and connecting local residents. Our professionals explained this phenomenon as follows:

> *"The new scenarios enriched the planning content based on the development directions of the current plan. The current desert park provides recreational opportunities for desert sports and related adventures. It provides professional Outward-Bound facilities and has a clear targeted customer market. However, it does limit the average number of visitors that will explore this region."*. (Professional 1)

> *"Compared with the current plan from the decision-makers alone, the other scenarios reminded us to add a theme to the desert sports park that is based on local, cultural resources. I like the idea of this region in village 1 mainly being focused on desert sports and recreation, with an integration of local Silk Roads cultural elements. Then we could make this desert park stand out from other desert recreation areas, using a unique theme. Additionally, the new ideas in other scenarios reminded us to improve the connections between the plan and local village representatives. The supporting area combined with local culture could provide board opportunities involving residents that benefit their life (such as the spaces for village markets and snack streets). This region could not only be a desert recreation destination or an outdoor training camp, but also a rustic luxury resort combining special Silk Roads cultural resources and desert scenery. "* (Professional 4)

In sum, our professional respondents pointed out some general limitations of decision-makers and, as a result, these new scenarios can help related planners incorporate knowledge from all available resources during planning to understand the existing environmental local, and social expectations for land use. As one professional put it:

> *"The newer scenarios bring in some elements that might inspire us about the expectations of local residents. A decision maker's knowledge can sometimes be limited in this area, and even an expert's skills can vary. Thus, local resident views can provide a localized, contextual knowledge aiding in long-term development."* (Professional 5)

## 6. Discussion

### 6.1. Scenarios Based on Development Preferences and Professional Planner Assessments of Local Sustainable Development

Our results indicate that the professional assessments of the different scenarios were largely in favour of combining local resident and decision-maker development preferences. In general, professionals found that many of the suggestions proposed by local residents were unrealistic because of cost—both in construction and in maintenance. As a result, scenario 1 (the scenario primarily guided by local resident preferences) received the lowest score, as ranked by professionals who offered more negative comments about this scenario in general. A major reason was that related facilities such as the artificial canal and the larger-scale historical buildings required correspondingly large investments. That said, four of the five professionals nevertheless found that local resident participation did add new and good ideas to each scenario. Consequently, professional respondents marked scenario 4 (the scenario where decision-makers' development preferences are combined with local ideas) as the most preferred alternative.

The arguments for giving scenario 4 the highest rank can be summarized in three points: (1) Scenario 4 has a more comprehensive functionality that combines both the cultural tourism and desert sports recreation aspects of the possible development plans; (2) Scenario 4 requires moderate investments as well as only limited construction volume; (3) Scenario 4 has better functional zoning solutions and more sustainable use of local natural resources and environmental opportunities.

Together, the four professionals asserted that scenario 1 (the scenario-based exclusively on local resident preferences) required the highest level of investment for building, and would also be the most expensive to maintain. There was an additional worry among the group regarding competition with other similar historical tourism centres already planned along the Silk Roads corridor region. The professionals found that scenario 2 (the scenario-based exclusively on decision-makers' development preferences) had some inherent limitations; in this case, the mono-functional land uses for the desert sports park in scenario 2 would mean more special desert sports functions than general recreation, which would appeal to a more limited group of tourists. For scenario 3, professionals expressed concern that, even though the planning of this scenario (local preferences combined with decision-makers' ideas) did, in fact, consider both the function of cultural tourism and desert recreation, the functional zoning plans in scenario 3 were unfortunately piecemeal and required higher construction investments. In short, the smaller scale of the desert recreation zones compared with scenario 4 would limit desert recreation activities, reduce service functions, and ultimately compromise tourists' overall experience of the town.

In general, scenarios 2 and 4 generally have higher ranks than scenarios 1 and 3. Professionals stated that their rankings were not influenced by their prior knowledge of the direction of desert sport park. In their collective view, a well-designed desert sports park indeed could be an attractive tourism feature. Further, in comparison with cultural tourism, they viewed the desert sport park as a novelty in the local area.

The fourth professional preferred scenario 2 to scenario 4, as he felt resident suggestions did not add value to the development and planning strategies. In the review of the professionals, professional number four also stated that their reason for ranking scenario 4 as the most suitable for sustainable development was because it has a clear focus on desert sports recreation, naturally highlighting the particular organic qualities of unique, local, and natural resources.

To sum up, even though our professional respondents thought that many of the suggestions from local residents were unrealistic, the majority of them nevertheless asserted that local resident participation contributed to a better planning solution when combined with current decision-maker plans (shown in scenario 1). For that reason, local resident participation has the potential to cement the success of an improved corridor-planning proposal for local sustainable development, benefitting both the local residents and the decision-makers involved.

### 6.2. The Difficulties of Involving Local Residents during Development and Planning Processes in China

In China, local urban and rural communities have already developed some participatory institutions [41], where currently a range of experiences with local residents' participation has been gained [29,42]. This shift could ultimately develop into an era of increased local resident participation in China.

While researching solutions, scholars have thus concluded that barriers to local residents participation and collaborative planning in China lie not only in its weak framework for environmental legislation, but also in part because of Chinese culture [30], the weakness of its planning system, and limited tradition of incorporating local resident values into planning practice [43,44]. Over the course of our study, we found that the existing gap between the differing views and education levels of our professional respondents, local decision-makers, and local residents also contributed to the difficulties of involving the general population during landscape corridor planning in China. We therefore discuss this condition from both a local resident and planner perspective as follows:

Our study used face-to-face interviews combined with local resident participatory GIS and group meetings for the purpose of communicating with local residents and collecting ideas and preferences concerning the future development of this part of the Silk Roads corridor. When local residents were asked about their preferences, we found that they more often had higher expectations of construction works, and that their development preferences were highly influenced by their general views of modernization and economic development on the whole (i.e., as seen via social media). For instance, the idea of developing a canal to provide artificial water scenery in the historical district from scenario one was an idea local resident participants mentioned they got from another cultural tourism town they saw on television. Clearly, local resident views of future developments cannot replace more trained, professional inputs on development opportunities and limitations.

During our communications process, the professionals saw themselves as key players, essential to the planning process and with the capacity to link existing resources with development opportunities and limitations. They were the experts with the technological skills necessary for the plan-making process and did not really recognize the need to explore, much less incorporate, local preferences and livelihoods during the process. As a result, their spatial and planning processes were viewed more as an exercise in technical engineering, than as a social activity that included collaboration with local stakeholders. This technical view of planning may explain why planners in China currently have such a low motivation to share their values and experiences with local residents.

Our data collection process revealed the current practical difficulties inherent in coordinating the steps it takes to consider local resident opinions during planning processes in China. These difficulties relate to the existing gaps between educated planners and "uneducated" residents, which can reduce a planners' motivation to incorporate local resident participation in their practices. To further develop sustainable participation practices, decision-makers must therefore address this gap, and local residents must on their side accept that not all their ideas and preferences are feasible from economic and technical points of view.

Educating planners, authorities, and residents is a prerequisite to more collaborative planning practices. Local resident participation in China may not only represent a better approach to formally allowing people to express their views, but it is also a process that promotes cooperation between planners and motivating them to share their experiences and knowledge with the local residents. This should also include raising local awareness about sustainable development options, strengths, and limitations. The planning process could then combine local resident opinion and high-level planning to better analyze the conditions of planning areas; better ascertain geological and socioeconomic strengths and weaknesses; and ultimately further explore development factors (i.e., opportunities and threats). The value of local resident participation is thus demonstrated as an enrichment mechanism for the current plan.

## 7. Conclusions

Landscape corridor planning is a complex process which requires close collaboration between various stakeholders to be efficient and innovative, concerning both the place-making and conflict-management dimensions of planning practice. In the current state of participatory landscape corridor planning, our study illustrates a practical way to involve public participation in LCP solutions that may also be applicable to other forms of spatial planning. Our study demonstrates the concrete differences that exist between planning solutions with and without local resident participation, thereby revealing how spatial conflicts might be reduced through collaborative processes. In addition, our results highlight the extent to which local resident participation indeed contributes to innovation, and professionals acknowledged that an enriched current corridor plan could contribute to local sustainable development in China. Further research is required to understand—and remove—the barriers to promoting local resident participation in Chinese spatial planning practice, including research on various forms of involvement, combined planning and educational processes, and ways to mobilize local knowledge and ideas during the planning process.

**Author Contributions:** Conceptualization, H.X.; Methodology, J.P., and T.P.; Software, H.X.; G.Z.; Validation, H.X. and J.P.; Investigation, H.X.; Data Curation, H.X.; Writing Original Draft Preparation, H.X. and J.P.; Writing Review & Editing, J.P. and T.P.; Visualization, H.X.; Supervision, J.P. and T.P.; Project Administration, H.X.

**Funding:** This paper is sponsored by the Chinese Scholar Council.

**Acknowledgments:** We are grateful to the great help and support of Prof. Xuesong Xi from China Agricultural University for providing the master plan and supporting the cooperation with the Municipality. The landscape architects and urban planners, Bingbing Zhang, Leng Gang, Xinya Bei, Yanli Tao also provide their input in this paper.

**Conflicts of Interest:** The authors declare no conflict of interest.

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
