# Peer review of "What Difference Does Public Participation Make? An Alternative Futures Assessment Based on the Development Preferences for Cultural Landscape Corridor Planning in the Silk Roads Area, China"

_sustainability, doi:10.3390/su11226525_

Round 1
Reviewer 1 Report
This is an interesting study that examines the impacts of public participation in landscape corridor planning in China where is often known as not a democratic country, especially for the political system. The methodology is well designed and the results provide useful insights for the future diffusion of participatory planning in China. Thus, I can say that this is acceptable after several revisions that mainly result from partially not understandable presentations. My comments are as follows:
1) This study targets landscape corridor planning while the distinct characteristics of it compared to common landscape planning or rural planning are not clearly stated. I want you to make clear if the study contributes only to landscape corridor planning or to more general (rural) planning.
2) Figure 4 shows important results while readers cannot easily understand this figure and the following explanations in p13. It is because the Kernel density hear maps do not have the location of the ten study villages while sentences mention each of the villages. Although it is shown in Figures 1 and 2, it is not comfortable to frequently go back to Fig. 1 or 2 from here.
3) Visualization of the four scenarios looks interesting and appealing. However, I could not understand them enough. Where do the enlarged areas show in the whole study area? From which viewpoint are they seen? Besides, what is drawn in the background picture most of which is hidden by the enlarged area? I also want to see the current (original) landscape for comparison.
4) I think you can mention more about how you asked experts for ranking the four scenarios. For example, what was the order of their considering the scenarios, how long it was taken for the assessment for each scenario or how much was each expert involved in the planning? If there are some biases in the degree of their involvement in the planning, it may affect the result. Besides, it might be natural that scenarios 2 and 4 have higher ranks if the experts already knew scenario 2 well through their planning experiences. If so, it should be considered in the discussion.
5) The sub-section 5.2 can be reconsidered. It contains new but general information on the situation in terms of public participation in planning in China. I felt it sudden. The first and a part of the second paragraphs can be moved to the introduction and you can emphasize the necessity of this study based on the current situation in China. The last paragraph is the only for the conclusion, so you can add a section for it wich name of 'Conclusion'.
Author Response
Thank you so much for your comments. Please find our point-by-point responses as the attached file.

Reviewer 2 Report
This is indeed a relevant, interesting, well structured and clearly written paper that merits being published in Sustainability. It does contribute with empirical evidence to the multiple benefits of participatory processes for enhancing landscape corridor planning. Furthermore, it does provide with such an evidence in a socio-political context, the People´s Republic of China, where top-down and centrally defined planning regimes are still prevalent.
Nonetheless, and before proceeding towards its publication, I would encourage the authors to perform a few (minor) edits and changes. These are:
i) Along the whole paper (actually since the abstract itself) emphasis is made on the planning context and process, but nonetheless, the reader might get the feeling that authors incurr in a certain excess in the use of this term, without much clarifying what specific meaning is adopted (Spatial Planning, Land-Use Planning, Planning as Policy, as Administrative Procedure or as Scientific Discipline, among others). The lack of clarity to which I am referring may be ilustrated by the following sentence in the abstact (lines 17-18): "The planning process is complex including many policy and planning issues". This can mean anything, and is the kind of statement that turns the text confusing at times. I would suggest the authors briefly define clearly what their interpretation of planning is, why and how it adapts to the Chinese context, and the implications for other contexts where planning might have a different meaning and significance.
ii) The second big question that arises is the feeling that the authors are constantly trying (and even perhaps claiming) to prove something (that participation enhances planning) that is already well grounded in both planning theory and practice. My understanding that they refer that this as something novel in the National context of China, right?. If so, this caveat should be made more clear. Also, what are the lessons of relevance (if at all) for other national and regional contexts (e.g. Europe) where this is already a widely accepted (and even implemented) challenge?.
iii) This remark is merely visual, but may affect legibility and thus potrntially detract interest from the readership. Figures 6, 7 and 8 (scenarios) contain information that is too small and thus tough to read. Even though it might be zoomed in in its digital version, the printed version might become difficult to read, and thus figures should be edited accordingly.
iv) Lastly, it would be great if the text can be reviewed by a native English speaker so that all minor spelling and expression errors are adequately tackled.
I look forward to receiving and reviewing, if needed, a revised and improved version.
With best regards.
Author Response
Many thanks for your comments. Please find our point-by-point responses as the attached file.
